# Spatial-Temporal Patterns of Ecosystem Services Supply-Demand and Influencing Factors: A Case Study of Resource-Based Cities in the Yellow River Basin, China

**DOI:** 10.3390/ijerph192316100

**Published:** 2022-12-01

**Authors:** Li Ming, Jiang Chang, Cheng Li, Yedong Chen, Cankun Li

**Affiliations:** 1School of Mechanics and Civil Engineering, China University of Mining and Technology, Xuzhou 221116, China; 2Jiangsu Collaborative Innovation Center for Building Energy Saving and Construct Technology, Jiangsu Vocational Institute of Architectural Technology, Xuzhou 221116, China; 3School of Architecture and Design, China University of Mining and Technology, Xuzhou 221116, China; 4Research Center for Transition Development and Rural Revitalization of Resource-Based Cities in China, China University of Mining and Technology, Xuzhou 221116, China

**Keywords:** ecosystem services, supply-demand relationship, driving factors, Yellow River Basin, resource-based cities

## Abstract

The aim of this study was to reveal the spatiotemporal pattern of the supply and demand of ecosystem services (ESs), as well as the significant driving factors for understanding the impact of human activities on the natural ecosystem. To provide a scientific basis for formulating regional sustainable development strategies that enhance human well-being, resource-based cities in the Yellow River Basin (YRB) were selected as the case study. The supply and demand of ecosystem services in these cities from 2000 to 2020 were measured. The spatiotemporal evolution of the supply-demand relationship was illustrated by taking its coordination degree. In addition, geographical detector and geographically weighted regression (GWR) models were applied to quantify the spatiotemporally varying effects of natural and socioeconomic factors on the ES supply--demand relationship. The results showed that resource-based cities in the YRB were experiencing expansion in supply and demand overall, but the supply-demand relationship tended to be tense. The northwest YRB had higher coordination values of supply-demand, while lower values were found in the southeast YRB. Moreover, the relationship between supply and demand was significantly affected by natural and socioeconomic factors, such as elevation, slope, precipitation, land-use type, population density, and gross domestic product (GDP) per land. Furthermore, the GWR model suggested that the effects of driving factors on the supply-demand relationship had notable spatial heterogeneity. The coordination of ES supply-demand in the resource-based cities of southeast YRB was mainly influenced by socioeconomic factors, while that of the west YRB was mainly influenced by natural factors. Our study suggested that it is necessary to enhance the awareness of environmental protection, pay attention to ecological restoration, and avoid unreasonable human disturbance to the ecosystem.

## 1. Introduction

The continuous improvement of human well-being is the main objective for social development and the vision of human society [1,2]. Former research on human well-being mainly focused on economics, sociology, and social sciences, while the contribution of the ecological environment to human well-being has not received enough attention [3,4]. However, the truth is that improving human well-being in a sustainable way is extremely difficult given the realities of ecological degradation, environmental pollution, and the loss of biodiversity [5]. The UN Millennium Ecosystem Assessment (MA) proposed a framework for linking ecosystem services and human well-being; 60% of global ecosystem services are declining, while the consumption of over 80% of ecosystem services is increasing, with human wellbeing directly or indirectly affected by these ecosystem services [6]. Although scholars have conveyed the concept of ecosystem services in a variety of ways for various study contexts and purposes, there is consensus on the basic meaning and connotation of ecosystem services (ESs) as the benefits that humans derive directly or indirectly from the natural world [7].

The study of the interactions and interlinkages between ESs is one of the core elements of ESs research. For the purpose of encouraging the sustainable management of natural capital, it is crucial to understand the linkages between different ecosystems [8]. The supply of ecosystem services (ESs supply) depends on the structure, processes, and functions of ESs, which exist objectively rather than according to human will [9]. The ESs supply is an indicator to measure the potential capacity of ecosystems [10]. Demand for ecosystem services (ESs demand) refers to the utility that human society obtains through the consumption of ecosystem services in order to meet its own needs for survival and development [11,12]. Early scholars focused on the ESs supply, ignoring the ESs demand by human societies [13]. As human demand for ESs affects the functioning of the whole ecosystem, the need for systematic management of ecosystem services can no longer be met by considering only the ESs supply [14]. Therefore, more scholars have become interested in the relationship between ESs supply and ESs demand [15], and they have conducted numerous theoretical [16], methodological [17], and practical studies [1]. Current research focuses on the supply and demand for services of a single ecosystem type, ranging from provision services [18] to regulating services [19] and cultural services [20]. These studies on ESs supply-demand range in scale from small watershed studies to urban [1], regional [21], and national [22] case studies. They mostly focused on ES supply-demand at one particular moment [23]. Fewer studies have been conducted on the evolution of ecologically fragile areas in watersheds where human-land conflicts are prominent. The ES supply-demand is linked to natural ecosystems and socioeconomic systems [24]. Research on the long-term evolution of ES supply-demand can help to strengthen ecosystem management and optimize resource allocation, thus, ensuring regional ecological security and sustainable socioeconomic development.

Most studies on the drivers of ESs are based on qualitative analysis and correlation analysis, mostly from the supply-side perspective [25,26]. In contrast, there are few studies quantifying the drivers of matching ESs supply and demand [23]. Moreover, the key influencing factors were mostly revealed by single methods, such as correlation analysis and ordinary multiple linear regression. One of the key challenges in ecosystem management is the lack of comprehensive quantitative studies on the spatiotemporal variability in the influence of various factors on supply and demand relations. It is difficult to reveal, in-depth, the mechanism of the evolution of supply-demand relations under the combined effect of multiple factors. In this study, geographical detector and geographically weighted regression (GWR) models [14] were used to explore the dynamic evolution of supply-demand relationships under the influence of natural and socioeconomic factors. This has importance for the effective management of regional ESs and related policy formulation.

The Yellow River Basin (YRB) is an important ecological barrier and economic area in China [27]. It is now facing the situation of a fragile ecological background, a high resource and environmental load, and insufficient economic development. The quality of development in the YRB needs to be improved. The YRB is rich in coal, oil, natural gas, and minerals, and more than 30 resource-based cities have been formed on the basis of resource extraction and processing [28]. They account for more than 50% of the total number of cities in the basin and about 30% of the total number of resource-based cities in China [29]. Resource-based cities are some of the regions with the highest concentration of conflicts among population, resources, and the environment. Their ecological background and socioeconomic characteristics have undergone profound changes under the combined influence of mining activities and urbanization [30]. The imbalance between ESs supply and demand in resource-based cities is a major challenge to China’s major national strategy of “ecological protection and high-quality development of the Yellow River Basin”. Therefore, resource-based cities in the YRB were chosen as the case study. By exploring the matching of ES supply-demand from 2000 to 2020, the spatiotemporal evolution of supply-demand was elucidated. This, in turn, revealed the influence of natural and socioeconomic factors on the evolution of ES supply-demand. This will provide a scientific basis for further optimizing the spatial development pattern, thereby improving the sustainability of regional ecosystems and promoting human well-being.

## 2. Materials and Methods

### 2.1. Study Area

Before entering the Bohai Sea, the Yellow River travels through nine provinces (autonomous regions), including Qinghai, Si-chuan, Gansu, Ningxia Hui Autonomous Region, Inner Mongolia, Shaanxi, Shanxi, Henan, and Shandong. The Yellow River Basin (YRB) is the most important birthplace of Chinese civilization, spanning the following three regions: east, central, and west, located between 96° and 119° E and between 32° and 42° N, with a total area of 795,000 km^2^ [31,32]. The YRB is comprised of 38 resource-based prefecture-level cities, according to the “Comprehensive Plan for the YRB (2012–2030)” and the “National Resource-Based City Sustainable Development Plan (2013–2020)”. Due to the difficulty in obtaining relevant data in the Aba Tibetan and Qiang Autonomous Prefecture, and the fact that most of the jurisdictions belong to the Yangtze River Basin, the remaining 37 resource-based cities were used as the sample in this study (Figure 1).

### 2.2. Data Sources

The data used in this study were as follows: (1) land-use data (2000, 2010, and 2020) from GlobeLand30 (http://www.globallandcover.com, accessed on 1 September 2021), a global land-cover dataset with a spatial resolution of 30 m; (2) topographic data with a spatial resolution of 30 m using the digital elevation model (DEM) from the Geospatial Data Cloud (http://www.gscloud.cn/, accessed on 1 September 2021), with a spatial resolution of 30 m; (3) meteorological data, from the spatially-interpolated dataset of the average annual temperature and annual precipitation provided by the China Meteorological Data Network accessed (http://data.cma.cn/, on 1 September 2021), with a spatial resolution of 1 km; (4) socioeconomic data of resource-based cities in the YRB, sourced from the *China Urban Statistical Yearbook*, *China County Statistical Yearbook*, and the statistical yearbooks of the 37 resource-based cities in the YRB.

### 2.3. Quantifying ESs Supply

The term “ESs supply” describes the capacity to offer ecosystem products and services in a particular location at a certain time. The evaluation of ESs was based on the “China Ecosystem Service Value Equivalent Factors Table” created by Gaodi Xie [33,34], where each form of land-use or land cover was given a particular equivalent factor value. The method has the advantage of high data accessibility, low data requirements, and clear evaluation representations. This study classified ESs into 11 categories of service functions, such as raw materials and food production, and accounted for the value of ESs per unit area of each land-use type, as follows in Equations (1) and (2):(1)$$E_{a}=\frac{1}{7}\sum_{i=1}^{n}\frac{P_{i}G_{i}}{Ai}$$
(2)$${\mathrm{ESV}}_{j}=\sum E_{a}\times e_{ij}\times M_{i}$$
where *E_a_* is the economic value providing the food production service function per unit area of the agro-ecosystem (CNY/hm^2^), *i* is the crop type, *P_i_* is the average price of *i* food crops in the study area (CNY/kg), *G_i_* is the total yield of *i* food crops (kg), *A* is the area under food crops (hm^2^), 1/7 means that the economic value provided by a natural ecosystem without human input is 1/7 of the economic value of the food production services provided by the existing unit area of farmland [35], *ESV_j_* is the total value of the *j* service function of the ecosystem (CNY/year), *e_ij_* is the equivalent coefficient of the *j* service function of the ecosystem of land-use type *i* relative to the unit of ecological services provided by the farmland ecosystem, and *M_i_* is the area of land-use type *i* (hm^2^/year). The total ESs supply in the study area can be obtained by summing the values of each ESs function for different ecosystems.

The ESs value equivalent factor is the potential capacity of an ecosystem to produce the relative size of its contribution to ESs. The value equivalent method by Gaodi Xie determines that the economic value of one ESs value equivalent factor is equal to 1/7 of the market value of the national average grain yield in China in that year [33]. Drawing on the value equivalent conversion method, the scale of value equivalents of ESs was determined in the study area. The value of ESs equivalents in the YRB was determined using a correction factor of 0.93 according to the ratio of grain production per unit area in the study area to the grain production per unit area nationwide; this correction factor was used to calculate the equivalence coefficients of all services. The average prices of grain crops in 2000, 2010, and 2020 were 1.14 CNY/kg, 1.81 CNY/kg, and 2.38 CNY/kg, respectively. In order to eliminate the effect of currency depreciation on price fluctuations, the average price of food crops in 2010 was calculated to be 1.71 CNY/kg, using an annual currency depreciation rate of 6.83% for the period 2000–2020. The correction coefficients (Table 1), from which the ESs supply provision in the study area was determined, were substituted to determine the equivalency coefficients of ESs per unit area for various land-use types in the YRB research region.

### 2.4. Quantifying ESs Demand

The ESs demand refers to the utility that human society obtains through the consumption of ESs to meet its own survival and development needs [36]. Three socioeconomic variables were used to calculate the ESs demandafter considering the factors influencing ESs change and the data’s availability, namely population density, the degree of land-use development, as in Equation (3), and GDP per land, as in Equation (4). The two indicators fluctuate greatly due to the considerable variations in population density and GDP per land in a select few extremely developed regions. In order to make subsequent analysis easier, the statistical approach of taking logarithms allows the characteristics of local sharp oscillations to be faintly descended into the calculations without changing the general distribution trend. Equations (3) and (4) are as follows:(3)$$x^{*}{}_{i1}=100\times \sum_{i=1}^{n}A_{i}P_{i}/A_{T}$$
(4)$$X=x_{i1}^{*}\times \mathrm{lg}\left(x_{i2}^{*}\right)\times \mathrm{lg}\left(x_{i3}^{*}\right)$$
where X represents the ESs demand in the evaluation unit, *x^*^_i_*_1_, *x^*^_i_*_2_, and *x^*^_i_*_3_ represent the degree of land-use development, population density, and GDP per land, respectively, *A_i_* is the area of land-use type *i*, *A_r_* is the total area of the study area, and *P_i_* is the parameter of the degree of land-use of different types; this value is obtained by the predecessors [37,38] combining the expert scoring method and matrix method (Table 2).

### 2.5. Analysis of Supply-Demand Pattern of ESs

#### 2.5.1. Supply-Demand Pattern of ESs

The ESs supply and demand were standardized by z-score using the above method of accounting for ESs supply and demand. The standardized ESs supply was used to characterize the ecological background on the *x*-axis and ecological demand on the *y*-axis, divided into four quadrants representing different types of ES supply-demand patterns, namely high supply/high demand, low supply/high demand, low supply/low demand, and high supply/low demand. The standardized method by *z*-score for ESs demand and ESs supply can be expressed as follows:(5)$$x=\frac{x_{i}-\bar{x}}{s}$$
(6)$$\bar{x}=\frac{1}{n}\sum_{i=1}^{n}x_{i}$$
(7)$$s=\sqrt{\frac{1}{n}\sum_{i=1}^{n}{\left(x_{i}-\bar{x}\right)}^{2}}$$
where *x* is the ESs supply and ESs demand after standardization of the evaluation unit, *x_i_* is the ESs supply and ESs demand of evaluation unit *i*, $\bar{x}$ is the mean value, *s* is the standard deviation, and *n* is the total number of evaluation units.

The quantitative analysis and evaluation of the state of supply-demand has important theoretical and practical implications for ecosystem management, regional ecological security, and sustainable human socioeconomic development [39]. The coordination degree of ESs supply and demand (ESDR) in this study relates the actual supply of ESs to human demand, which can be used to reveal the nature of surpluses or deficits, as calculated by the following formula:(8)$$ESDR=\frac{S-D}{\left(S_{\max}+D_{\max}\right)/2}$$
where *S* and *D* refer to the supply and demand after the standardized method by *z*-score, respectively; *S*_max_ is the highest assessed ESs supply in the assessment area, and *D*_max_ is the highest assessed ESs demand in the assessment area. Here, ESDR > 0 indicates an oversupply of ESs, ESDR = 0 indicates that supply and demand are in balance, and ESDR < 0 indicates an undersupply of ESs.

#### 2.5.2. Spatial Autocorrelation Analysis

Spatial autocorrelation analysis is a statistical method used to measure the spatial distribution characteristics of physical or ecological variables and the degree of their influence on the domain [40]. In this study, the global autocorrelation of the ES supply-demand degree was examined using Moran’s I value. The value range of Moran’s I is −1 to 1, where a positive value indicates a positive spatial correlation (a larger value indicates a larger spatial correlation), a negative value indicates a negative spatial correlation (a smaller value indicates a larger spatial heterogeneity), and a zero value indicates a random spatial distribution. Local indicators of spatial association (LISAs) were used to identify statistically significant high/high, low/low, high/low, low/low, and not significant spatial clustering patterns. Global and local spatial autocorrelation analysis can better describe the relationships between geographic events, measure the degree of aggregation or dispersion between the attributes of supply and demand spatial elements, and reveal the evolution of spatial clustering patterns that match the supply and demand for ESs.

### 2.6. The Driving Factors of Supply-Demand Pattern of ESs

#### 2.6.1. Geographical Detector Model

In this study, the geographical detector model proposed by Wang Jinfeng was used to analyze the factors influencing the supply and demand [41,42]. This model is a statistical method that explores spatial heterogeneity and reveals the combined effects of multiple factors driving it. In this paper, the factor detection and interaction detection modules were used to identify the influences of natural and socioeconomic factors on ES supply-demand and to explain the interactions of factors influencing the coordination degree of ES supply-demand to enhance the spatial delivery of services [43].

(1) Factor detection is achieved by comparing whether an influencing factor is spatially significantly consistent with changes in the coordination degree of ESs supply-demand. If they are consistent, then such a factor is determinative of the spatial variation in the coordination degree of supply and demand. Equation (9) is as follows:(9)$$q=1-\frac{1}{N\sigma^{2}}\sum_{i=1}^{m}n\sigma_{i}^{2}$$
where q is the effect of the influencing factor on the degree of supply and demand coordination, with a value range from 0 to 1, with larger values indicating a stronger explanatory power of the influencing factor. Here, *N* and *n* are the number of factors in layer *i* and the whole region, respectively, and *σ_i_*^2^ and *σ*^2^ are the variance of the coordination degree of supply and demand in layer *i* and the whole region, respectively.

(2) Interaction detection can quantitatively characterize the relationship between the effects of two influencing factors on the coordination degree of supply and demand. Firstly, the q(X1) and q(X2) values are calculated for a single factor. Secondly, q(X1∩X2) values are calculated for the interaction between two factors, and then q(X1∩X2) is compared with q(X1) and q(X2) to determine the relationship between the two factors.

#### 2.6.2. Geographically Weighted Regression Model

By generating a local regression equation at each point on a spatial scale, the geographically weighted regression (GWR) model investigates the spatial variability of a research object at a specific scale and the associated drivers. The GWR is an extension of the conventional linear regression model that considers the local impacts of the spatial object [44]. It is calculated as follows:(10)$$y_{i}=\beta_{i}(u_{i},v_{i})+\sum_{i=1}^{k}\beta_{j}\left(u_{i},v_{i}\right)x_{ij}+\varepsilon_{i}$$
where *y_i_* is the value of the dependent variable at the *i* sample point, (*u_i_, v_i_*) is the spatial geographic location coordinates of the *i* sample point, *β_j_* (*u_i_*, *v_i_*) is the *j* regression parameter at the *i* sample point, *x_ij_* is the value of the *j* independent variable at the *i* sample point, and *ε_i_* is the random error term. In this study, the spatial weight matrix is determined by a Gaussian function, while the optimal bandwidth is obtained by the Akaike information criterion (AIC) method, where the bandwidth is optimal when the AIC value is minimal. We use the GWR module in the ArcGIS 10.6 spatial statistics tool for the calculations.

## 3. Results

### 3.1. Spatiotemporal Characteristics of ESs Supply and Demand

Using the ecosystem service value equivalent and the demand accounting method, the total supply and demand of resource-based cities in the YRB in 2000, 2010, and 2020 were estimated. Taking the county as the basic space, the ArcGIS spatial analysis module was applied to output the estimation result unit, so that we could obtain the spatial and temporal distribution patterns of the supply and demand of resource-based cities in the YRB.

#### 3.1.1. ESs Supply

According to the land-use data, the ESs supply values of resource-based cities in the Yellow River Basin from 2000 to 2020 were calculated and their changes were analyzed. As shown in Table 3, the supply of ESs in the resource-based cities of the YRB had increased from 2000 to 2020, and the largest increase in water regulation function is 52.17%. This phenomenon was directly related to the implementation of ecological protection policies in the YRB in China. Compared with 2000, the total supply increased by 263.193 billion CNY or 41.92%, but the growth in supply from 2010 to 2020 was significantly slower than during the previous decade. In the past 20 years, the increase in supply in Dongying City was the most obvious, whereas the supply in the Jili District of Luoyang and the Cheng District of Yangquan decreased; both are located in areas with a concentration of resource-based cities in the middle reaches of the Yellow River.

Figure 2 shows that the spatiotemporal pattern of ESs supply in resource-based cities located in the YRB from 2000 to 2020 showed a more obvious regional heterogeneity, i.e., low in the southeast and high in the northwest. The total supply was higher in Erdos, Baotou, Zhangye, and Qingyang. The ESs supply was high in areas located in the upper reaches of the Yellow River. The highest multiyear average value of total supply of ecosystem services was 33.462 billion CNY in Ordos Ertok Banner. The lower total supply was mainly distributed in the more socioeconomically developed areas of the YRB, such as Tai’an, Jining, and Luoyang. Xigong District in Luoyang City had the lowest average total supply of 0.1 billion CNY.

#### 3.1.2. ESs Demand

The ESs demand index for resource-based cities in the Yellow River Basin between 2000 and 2020 was divided into six levels in ArcGIS (as shown in Figure 3). The ESs demand in resource-based cities in the YRB increased along with time. Compared with 2000, the high-demand area in 2020 increased by 51.87%. The growth of ESs demand varied according to region. The change in demand was inconspicuous in Erdos City, Zhangye City, and Wuwei City, which are located in the upper reaches of the Yellow River. However, the demand grew more quickly in Yan’an City and Weinan City in the middle and lower reaches of the Yellow River.

In terms of spatial distribution, the spatiotemporal pattern of ESs demand in resource-based cities in the YRB changed little over the past 20 years, showing a low trend in the northwest and a high trend in the southeast. The areas with high ESs demand were mainly in the eastern regions of the YRB, such as Jining, Zibo, and Luoyang, while the low-demand areas were mainly in the western regions, such as Zhangye, Wuwei, and Qingyang. The characteristics of the pattern of distribution of ESs demand were consistent with the characteristics of the distribution of the degree of economic development. The highest average ESs demand was recorded in Zibo, Shandong Province, whereas the lowest was recorded in Zhangye, Gansu Province (507.13 and 34.14, respectively). In addition, over the past 20 years, the ESs demand showed a clear “southeast-northwest” spread in growth pattern.

### 3.2. Spatiotemporal Pattern Analysis of ESs Supply-Demand

#### 3.2.1. ESs Supply-Demand Patterns

As shown in Figure 4, the relationship between ESs supply and ESs demand in resource-based cities in the YRB is studied using the quadrant method. The degree of ES supply-demand in resource-based cities in the YRB decreased along with time. In 2000, there were no areas in Quadrant I, whereas most areas were in Quadrants II (low supply/high demand; 118 counties) and III (low supply/low demand; 106 counties), and a small number of areas were in Quadrant IV (high supply/low demand; 89 counties). In 2010, there was one area in Quadrant I, while the number of counties in Quadrant II increased to 121. However, the number of counties in Quadrant III and Quadrant IV fell to 103 and 89, respectively. Furthermore, there was a tendency of the supply-demand situation to worsen in 2020 compared to 2000. The number of counties in Quadrant I increased to 2, whereas it fell to 102 and 88 in Quadrants III and IV, respectively. The resource-based cities in the YRB were facing serious ecological risks as a result of the overall trend of decreasing the degree of ES supply-demand.

The spatial and temporal balance of ESs supplyand demand can be reflected by the coordination degree of EG supply and demand. The resource-based cities in the YRB were separated into five types of area using ArcGIS’s isometric division algorithm. In the resource-based cities of the YRB, the distribution of ESs supply and demand was unbalanced. There were situations of both supply exceeding demand and demand exceeding supply. Figure 5 shows that the pattern of ES supply-demand in resource-based cities in the YRB from 2000 to 2020 changed little, despite significant changes in some years. Comparing the ES supply-demand balance between 2000 and 2020, we found a significant increase in in the distribution range of short supply between 2000 and 2020. This implies that these areas were exposed to higher risk of ES supply-demand imbalance, increasing the eco-environmental pressure.

The degree of supply-demand coordination was spatially distributed with a low value in the southeast and a high value in the YRB of the northwest. The northwest region of the YRB showed a trend of supply exceeding demand, while the southeast region of YRB showed a trend of demand exceeding supply. Furthermore, the imbalance gradually became more serious, leading to the expansion of the undersupplied area from the southeast to the northwest YRB. Specifically, the number of mismatched supply and demand areas with a supply-demand coordination degree of less than 0 increased year by year, from 150 in 2000 to 153 in 2020. Furthermore, the areas were mainly concentrated in eastern regions with a high socioeconomic level, such as Shandong and Henan. The serious mismatch in supply-demand across different areas (coordination degree <−0.15) showed a trend of expansion first and then contraction, with the characteristics of a small area and dispersion. The large area of built-up land in these areas and the low coordination degree of ES supply-demand make the ESs supply unable to meet the growth in human demand, which manifests itself as the ESs deficit. The areas where supply exceeded demand were large (matching degree greater than 0) and mainly concentrated in the northwest of the study area, such as in Gansu Province and the Inner Mongolia Autonomous Region, belonging to the upper reaches of the Yellow River Basin, which are rich in ecological resources and relatively abundant in ESs supply.

#### 3.2.2. Spatial Autocorrelation of ESs Supply-Demand Pattern

The global spatial autocorrelation of ES supply-demand coordination degree in resource-based cities in the YRB in 2000, 2010, and 2020 was calculated using ArcGIS10.6 software. The Moran’s I values were 0.4365, 0.4068, and 0.4279, respectively. The *z*-values were 19.2285, 17.9260, and 18.8577, respectively, and all measurements passed the 99% confidence test. The Moran’s I values were all positive, indicating a significant positive spatial correlation between ESs supply and ESs demand in resource-based cities in the YRB. The change in Moran’s I values shows that the spatial clustering was significant, and the change in clustering was phased. The Moran’s I values decreased significantly from 2000 to 2010, indicating that the spatial clustering of the supply-demand balance weakened. However, the Moran’s I values increased each year from 2010 to 2020, indicating an increasingly significant spatial clustering of the supply-demand balance.

To further explore the spatiotemporal characteristics of the balance between ESs supply and ESs demand in resource-based cities in the YRB, the local spatial autocorrelation measure was conducted. The LISAs method was used to analyze the coordination degree of ES supply-demand from 2000-2020, which revealed five spatial distributions of clusters, namely high/high, low/low, low/high, high/low, and not significant. As shown in Figure 6, the spatial heterogeneity of the coordination degree of ES supply-demand in resource-based cities in the YRB was large, but the spatial distribution pattern was less variable. The study area was dominated by four types, namely low/low, high/low, high/high, and not significant clusters, whereas there was only one low/high cluster, which was the Kangbashi District of Ordos City. Spatially, the spatial layout of the coordination degree of ES supply-demand in the study area formed three clusters, namely a high/high cluster around Erdos City and Yulin City in the northwest YRB, a high/low cluster in the central region connecting Changzhi City, Jincheng City, and Luoyang City, and a low/low cluster in the southeast YRB with Jiaozuo City and Tai’an City as the main locations. Areas of low/low clusters were concentrated in the east, where socioeconomic development is faster, while areas of high/high clusters were concentrated in the north-west, where the ecological background is better. From 2000 to 2020, the number of low/low clusters rose from 84 to 88 and then fell back to 84, while the number of high/high clusters decreased from 37 to 33, indicating the deterioration of the ES supply-demand relationship between the resource-based cities in the YRB and the increasing ecological risk.

### 3.3. Influencing Factors for the Relationship between ESs Supply and Demand

#### 3.3.1. Geographical Detector Model

According to the results of the geographical detector model’s factor detection (shown in Table 4 and Figure 7), a combination of socioeconomic and natural factors were responsible for the spatial differences in the distribution of the degree of supply-demand coordination of ESs in resource-based cities in the YRB. The ranking of the average explanatory power of each dimension in the 3 years was population density > proportion of land used for construction > GDP per land > elevation > proportion of agricultural land > temperature > slope > precipitation > proportion of woodland > stages of city development. All the influencing factors passed the significance test, indicating that these indicators had a significant direct impact on the differences in the coordination degree of ES supply-demand in the study area. The contribution rates of topographic factors, such as elevation and temperature, to the coordination degree of supply and demand were higher, reaching 0.556305 and 0.366925, respectively, with explanatory powers >30%. Therefore, elevation and temperature were the main natural factors affecting the coordination degree of ES supply-demand. The economic and social factors with more than 70% explanatory power included population density and construction land ratio, reaching 0.760138 and 0.732738, respectively, which contributed greatly to the coordination degree, followed by GDP per land, reaching 0.639733. Therefore, the contribution of economic and social factors to the supply-demand pattern in the study area was significantly higher than that of natural factors.

As shown in Table 5, revealing the interaction detection for the interrelationships among driving factors, there was an interaction between natural and socioeconomic factors affecting the spatial variation in the ES supply-demand in resource-based cities in the YRB. Any two factors acting together had a higher impact than any one item acting alone, and the interaction between factors increased nonlinearly. The interactions of the proportion of agricultural land with the proportion of land used for construction, the GDP per land with the population density, and the population density with the proportion of agricultural land had a strong influence on the spatial variation of the coordination degree of supply-demand, with q-values reaching 0.856372, 0.851199, and 0.842665, respectively. This indicates that socioeconomic factors, particularly land-use type, had a strong influence on the coordination degree of ES supply-demand. The next strongest interaction effects on the spatial variation in the coordination degree of ES supply-demand were population density and slope (q = 0.818894), population density and proportion of woodland (q = 0.80921), population density and proportion of land used for construction (q = 0.806254), proportion of land used for construction and elevation (q = 0.802079), and proportion of land used for construction and precipitation (q = 0.800414), with the explanatory power of all these interactions exceeding 80%.

#### 3.3.2. Geographically Weighted Regression (GWR) Model

In order to further explore the spatial differences in the impact of key determinants of the supply-demand relationship’s compatibility, the classical algorithm for calculating the spatial heterogeneity distribution of geographic elements, i.e., the GWR model (with spatial reliability after the introduction of spatial weights), was selected for analysis. To reduce errors in the GWR results, it was necessary to diagnose the collinearity of independent variables and eliminate any such cases [14]. The results of this study’s usage of the variance inflation factor (VIF) and significance test to check for collinearity between the various variables are shown in Table 6. All factors that passed the VIF test but did not pass the significance test were excluded (temperature and stage of city development). Accordingly, the elevation, slope, precipitation, proportion of land used for construction, proportion of woodland, proportion of agricultural land, population density, and GDP per land were identified as the explanatory variables in the GWR model for the impact factor analysis.

Through global and local autocorrelation analysis, significant spatial autocorrelation characteristics were present in the ES supply-demand matching relationship in resource-based cities in the YRB. In this study, the eight indicators screened by ordinary least squares (OLS) were regressed using the GWR model, and the fixed Gaussian function was selected as the kernel function. The AICc method was used for the validation of optimal bands. The adjusted goodness of fit (*R*^2^) of the GWR model was much better than the OLS model, compared to the results of the two models, while the AIC values were significantly decreased (as shown in Table 7). This showed that the GWR model outperformed the OLS model and was better able to describe how various factors affected the supply-demand matching relationship’s spatial distribution. Compared to the global effect analysis of impact factors, revealing the spatial heterogeneity of key influencing factors and their roles serves as a significant guide for ecosystem management and coordinated regional development. In summary, the GWR model was used to provide insight into the mechanisms influencing the ES supply-demand relationship in resource-based cities in the YRB.

The results of the geographically weighted regression coefficient analysis (Figure 8) showed that the spatial distribution of influencing factors was localized and nonuniform. This showed that there was significant spatial nonstationarity and that the same factor had varying degrees of influence on the ES supply-demand at various spatial locations. A positive regression coefficient denotes that an increase in the influencing factor increased the likelihood of the coordination degree of ES supply-demand. A negative regression coefficient denotes that likelihood decreased with the increase in the influencing factor [45,46].

The influence of natural factors, such as elevation and slope, on the matching relationship between ESs supply and ESs demand showed little change in spatial pattern over the 20 year period. From the northwest to the southeast of the YRB, the influence of precipitation exhibited an increasing tendency. The regression coefficients for elevation were all less than 0, indicating that elevation had a negative effect on the matching of ES supply-demand. In terms of spatial distribution, the high-value areas of the regression coefficients were mainly concentrated in the Inner Mongolia Autonomous Region, Yangquan City, and Luoyang City, whereas the low-value areas were concentrated in Shandong Province. This indicated that elevation had a strong influence on the former regions but a weak influence on the latter. The regression coefficients of slope ranged from −3.622 to 1.394 in 2000, from −3.791 to 1.355 in 2010, and from −3.928 to 1.379 in 2020, indicated that both positive and negative effects of slope existed on the coordination degree of ES supply-demand. The high-value areas of the regression coefficients were mainly concentrated in Linfen, Pingliang, and Zhangye, decreasing toward the northwest and east YRB. Precipitation mainly had a positive effect on the coordination degree of ES supply-demand, whereas only the areas around Qingyang City, Jincheng City, and Tai’an City revealed a negative effect, gradually decreasing from 2000 to 2020.

The regression coefficients of the socioeconomic factors, including proportion of land used for construction, population density, and GDP per land were all less than 0, meaning that all three variables had a negative impact on the degree of ES supply-demand coordination. The proportions of woodland and agricultural land had effects on the supply-demand coordination degree, both positively and negatively. The ratio of agricultural land mainly had a negative effect, with only a few areas revealing a positive effect, i.e., Tai’an and Pingdingshan. The impact of the proportion of woodland was predominantly positive, with most of the areas revealing a negative effect concentrated in the Inner Mongolia Autonomous Region. The results show that the proportion of land used for construction and GDP per land had similar effects on the matching relationship between ES supply-demand. The high-value areas of the regression coefficient gradually expanded from east to west over time. The high-value areas of the regression coefficient of population density were mainly located in the northeast YRB, while the low-value areas were located in the west YRB, decreasing along this gradient. However, from 2000 to 2020, the number of low-value areas expanded steadily, while the high-value areas contracted significantly. In 2020, only a few high-value areas remained, i.e., in Datong, Yangquan, and Puyang.

## 4. Discussion

### 4.1. Calculation and Analysis of ESs Supply and Demand

In this study, the characteristics of spatiotemporal variation in ESs supply were derived, which can be used to assess the level of ESs supply and whether it has improved or declined in the counties of the resource-based cities in the YRB. The pattern of supply distribution in the study results was consistent with the distribution characteristics of vegetation cover, and correlated with the distribution of the Yellow River water system. The areas with a high value of ESs supply were mostly located in the upper reaches of the Yellow River and were mostly influenced by the Yellow River water system. The higher supply in these areas may be related to the better ecological background, high water supply, rich vegetation, better ecological functions, and sparse population. Areas with low values of ESs supply were mainly concentrated in the socioeconomically developed areas of the YRB, where urban construction land is predominant. The resource-based cities in the YRB saw a high increase in ESs provisioning capacity over the 20 years period, which was directly related to the implementation of ecological protection policies in the YRB in China.

In this study, we selected three indicators, namely land, population, and economy, to reflect the ESs demand in resource-based cities in the YRB. The spatial distribution of land utilization degree, economic development, and demographic status was consistent. Areas with a high demand for environmental resources tended to be in areas with large populations and developed economies, which were concentrated in the southeast YRB and in the middle and lower reaches of the Yellow River region, with greater regional development. The region’s high economic level and attractiveness to the surrounding population, with increasing urbanization levels and expanding areas of land for urban construction, led to growing ESs demand. Most of the northwestern YRB lacks dynamism in the economy, resulting in a constant exodus of the population and a slower rise in demand. At the same time, previous studies have shown that cities in the central (Shanxi, Shaanxi) and eastern regions (Shandong) have a stronger economic diffusion capacity [47]. The advantages of cities in resources, technology and labor force should be systematically infiltrated into the surrounding areas and larger areas to improve the economic development level and ability of these areas, thus, increasing ESs demand.

### 4.2. Analysis of the Spatiotemporal Evolution of ESs Supply-Demand Pattern

The relationship between ES supply-demand is primarily concerned with the coordination of human society and ecosystems [48]. Regional development facilitates an increasement in ESs demand from human society, which can be met by a greater supply capacity of ecosystems. Recognizing the relationship between ESs supply and demand is a basic prerequisite for resolving the man-land contradiction and achieving regional sustainable development.

Due to the multiple units of measurement used for supply and demand, the spatial relationship between the biophysical supply of ecosystems and the ESs demand from human society is unclear, which is a significant difficulty in the area. To address this issue, this study introduced an ES supply-demand accounting approach to analyze the spatial and temporal characteristics of the ES supply-demand relationships in resource-based cities in the YRB. In order to explore the relationship between ESs supply and ESs demand, a nonlinear curve fitting analysis was carried out using Origin for ESs supply and ESs demand from 2000 to 2020 as shown in Figure 9. The analysis results yielded the relationship expressed in Equation (11), as follows:(11)$$y=-0.84+0.09e^{-5.86x}$$
where *x* is the ESs supply, and *y* is the ESs demand, confirming the significant relationship between ESs supply and ESs demand.

The ESs supply and demand were spatially heterogeneous due to the physical geography and regional differences in socioeconomic development. However, the relationships were largely uncoordinated in most regions. Most of the areas where demand exceeded supply were concentrated in the southeast YRB, where the socioeconomic level is high. The balance between ESs supply and demand in these areas is at risk of becoming imbalanced. The flat topography of these areas, density of the population, production and trade activities, more significant population growth and urban expansion, and massive crowding out of ecological land by construction land have resulted in human demand greatly exceeding ESs supply. Areas with an oversupply of ESs were concentrated in the northwest YRB with a better ecological background. These areas were in the upper reaches of the Yellow River, with better developed ecological functions, a greater ESs supply, and a sparse population. However, China’s economic development strategy put forward the policy of supporting the developed areas in the east to support the backward areas in the west, so as to bring capital investment, talent investment, industrial investment, and other resources to the western region with a backward economic foundation, and to improve the infrastructure construction and industrial development mode in the western region. This policy has led to an increase in the rate of economic development in the west, an increase in the area of land used for construction, and a consequent increase in population. This has caused the area of high regression coefficients for economic factors to expand from east to west over time.

The pattern of the ES supply-demand balance was remarkably consistent over time. Although the supply-demand relationship was uncoordinated and tended to deteriorate overall, the degree of coordination improved between 2010 and 2020. This indicates that, since 2010, with the support of national policies, such as closing mines, returning farmland to forest, returning fields to lakes, afforestation, and ecological transfer payments, the ecological lands, such as water, wetlands, and woodlands in the YRB, have been effectively protected [49]. However, the balance between ESs supply and demand in resource-based cities in the east remains at risk of becoming unbalanced. The distribution range of areas with a short supply is increasing year by year. This suggests that further efforts are needed to improve the situation.

### 4.3. Analysis of the Drivers Influencing ESs Supply and Demand Patterns

In this study, the spatial characteristics of the factors influencing ESs supply and demand in the resource-based cities of the YBR were obtained using the geographical detector model. The contribution of socioeconomic factors was significantly higher than that of natural factors, with the interaction between socioeconomic factors being especially prevalent. The degree of supply and demand matching is reflected by the supply and demand index. Urban areas with high population levels in the resource-based cities of the YRB had higher levels of demand for almost all types of ESs, whereas areas with low population levels, such as the Inner Mongolia Autonomous Region, tended to have lower levels of demand. At the same time, population concentration is usually accompanied by the expansion of artificial land surfaces with high ESs demand and low ESs supply, whereas the occupation and destruction of natural ecosystems (e.g., woodlands, grasslands, and wetlands) is accompanied by low ESs demand and high ESs supply.

To further investigate the spatial variation of key influencing factors on the supply-demand relationship, the GWR model was used to analyze the spatial heterogeneity of the influencing factors in different regions. The key factors influencing the relationship between ESs supply and demand in resource-based cities in the YRB were elevation, slope, precipitation, proportion of land used for construction, proportion of woodland, proportion of agricultural land, population density, and GDP per land. Elevation, proportion of land used for construction, population density, and GDP per land exhibited significantly negative effects. Simultaneous positive and negative effects of slope, precipitation, proportion of woodland, and proportion of agricultural land existed on the matching relationship between supply and demand. In terms of spatial distribution, the absolute values of the regression coefficients of socioeconomic impact factors, such as GDP, population density, and the proportion of land used for construction, were greater in the eastern YRB than in the western YRB, but the high-value areas of socioeconomic impact factors tend to expand to the west. The natural impact factors, such as elevation, slope, and precipitation, were more influential in the western YRB than in the eastern YRB. The western region, located in the upper Yellow River region, is an important green ecological barrier with high vegetation cover and low population. The ESs demand is, therefore, lower in these regions compared to the eastern regions. Therefore, for ESs management in the future, different protection approaches should be considered. A method of exploiting soil and water resources that is compatible with the natural conditions and socioeconomic development level of the region should be chosen. In reducing the disturbance of ecosystems by human activities, greater attention should be paid to reducing the negative impacts of climate change on ecosystem services to obtain a stable ESs supply. On the contrary, the socioeconomic level of the eastern region is greater, with more significant population growth and urban expansion, resulting in a greater disturbance of ESs supply than in the western YRB. Therefore, the impact of socioeconomic development on the coordination degree of ES supply-demand in the east is more pronounced than in the west. In the future, sustainable management of the region should focus on adjusting the structure of demand while reducing human interference. Ecosystem restoration and rehabilitation should be promoted using artificial measures as a complement to achieve a sustainable equilibrium between supply and demand for ESs.

### 4.4. Limitations

There were some limitations to this study. On the one hand, with the aim of reflecting the level of ESs supply and demand, we selected representative ESs indicators that are critical to human well-being. However, the indicators for measuring regional ESs may be richer and more diverse due to the different physical geographic and socioeconomic characteristics of different regions. Hence, a subjective element can be added to the evaluation system in the future. On the other hand, ESs can be generated at different spatiotemporal scales. Differences in the ecological background and socioeconomic conditions at different scales can cause variability in the types, levels, spatial characteristics, and other aspects of ES supply-demand. This study examined the relationship between ESs supply and demand at the county scale. In subsequent studies, ESs supply and demand relationships can be studied at multiple scales to reveal their heterogeneity, with the aim of developing more targeted and implementable regional ecosystem management strategies.

## 5. Conclusions

This study took the resource-based cities in the YRB as the study area, using multisource data from 2000, 2010, and 2020 to measure the ESs supply and demand and determine the supply-demand degree of coordination. We first used a geographical detector model to explore the main factors influencing the coordination degree of ES supply-demand in resource-based cities in the YRB. Then, the GWR model was used to reveal the extent and mechanism of their influence on the supply-demand match at spatiotemporal scales. The results showed that the total ESs supply and ESs demand gradually increased over the past 20 years with the increase in economic and social development and ecological conservation awareness. In terms of spatial distribution, the relationship between supply and demand was spatially heterogeneous, but the relationship was largely uncoordinated in most regions. Most of the areas where demand exceeded supply were concentrated in the southeast YRB, where the socioeconomic level is higher. Most of the areas where supply exceeded demand were located in the upper Yellow River region, where the ecological background is better. The coordination degree of ES supply-demand in the southeast was mainly influenced by socioeconomic factors, while that in the west was mainly influenced by natural factors.

Although the ecological protection and high-quality development of the YRB have achieved positive results in recent years, resource-based cities in the YRB are facing serious ecological problems, such as surface subsidence, a reduction in agricultural land, land degradation, and soil erosion. The stagnation of green-oriented transformation in the resource-based cities of the YRB will inevitably affect the sustainable development in the YRB. Resource-based cities of the YRB where ESs demand exceeded supply were mainly due to the high level of socioeconomic development and significant urban sprawl. Therefore, we should strike a balance between economic development and ecological protection. The existing ecological space should be protected in strict accordance with the ecological red line by promoting intensive land-use and rational planning, as well as the restoration of ecological space, in order to reduce the imbalance between ESs supply and demand. The areas with surplus ESs in the resource-based cities of the YRB were mainly concentrated in the upper Yellow River region in the west. It is characterized by high levels of ESs supply and low levels of socioeconomic development. Policies that are conducive to population growth should be promoted. Policy should be planned from the following three aspects: socioeconomic development, ecological protection, water-land utilization. A protective approach for water-land utilization can be chosen for future economic development. The optimal allocation of resources is achieved by controlling the total demand for land and water resources, enhancing resource supply ability and maximizing their potential, thus, improving environment and the resource carrying capacity of the region. Green industries can also be introduced as appropriate to promote socioeconomic development. Moreover, an adequate monitoring and early warning mechanism for ecological safety should be established. We should also focus on ecological protection and improve ecological efficiency. Furthermore, we need to avoid pressure on the supply and demand balance of regional ecosystems due to the intensive anthropogenic disturbances.

## Figures and Tables

**Figure 1 ijerph-19-16100-f001:**
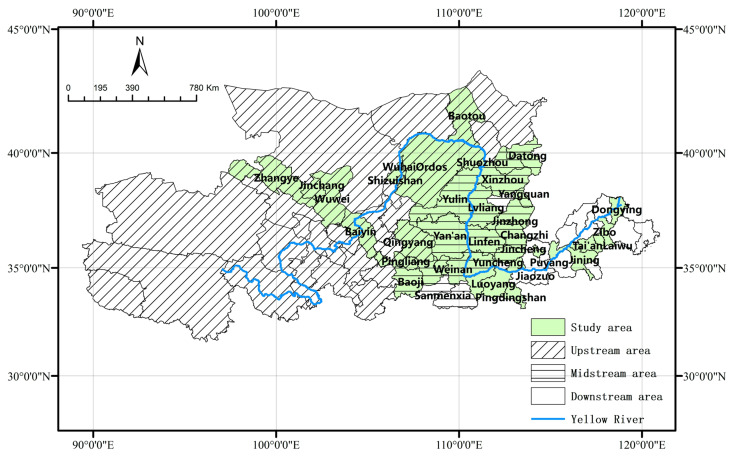
Study area.

**Figure 2 ijerph-19-16100-f002:**
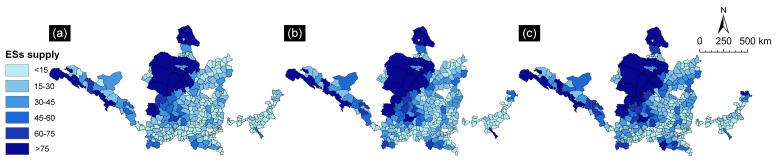
Spatial distribution map of ESs supply in resource-based cities in the YRB from 2000 to 2020. (**a**) 2000; (**b**) 2010; (**c**) 2020.

**Figure 3 ijerph-19-16100-f003:**
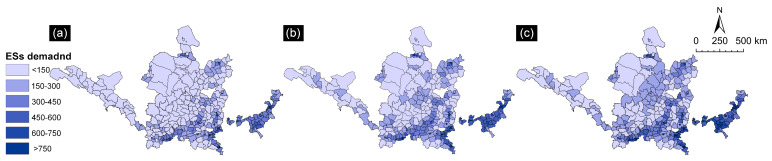
Spatial distribution map of ESs demand in resource-based cities in the YRB from 2000 to 2020. (**a**) 2000; (**b**) 2010; (**c**) 2020.

**Figure 4 ijerph-19-16100-f004:**
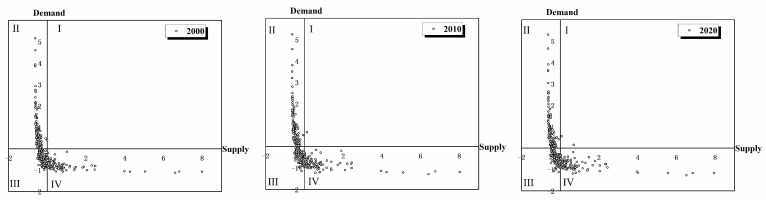
Correlation analysis of ES supply-demand in resource-based cities in the YRB from 2000 to 2020.

**Figure 5 ijerph-19-16100-f005:**
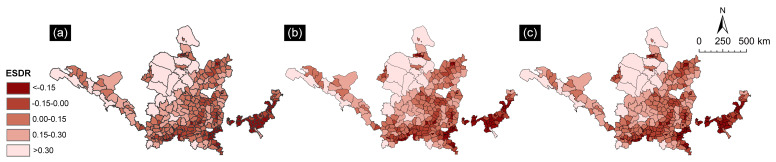
Coordination degree of ES supply-demand in resource-based cities in the YRB from 2000 to 2020. (**a**) 2000; (**b**) 2010; (**c**) 2020.

**Figure 6 ijerph-19-16100-f006:**
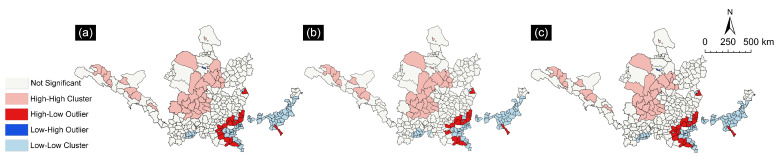
Local autocorrelation map of the coordination degree of the ES supply-demand of resource-based cities in the YRB. (**a**) 2000; (**b**) 2010; (**c**) 2020.

**Figure 7 ijerph-19-16100-f007:**
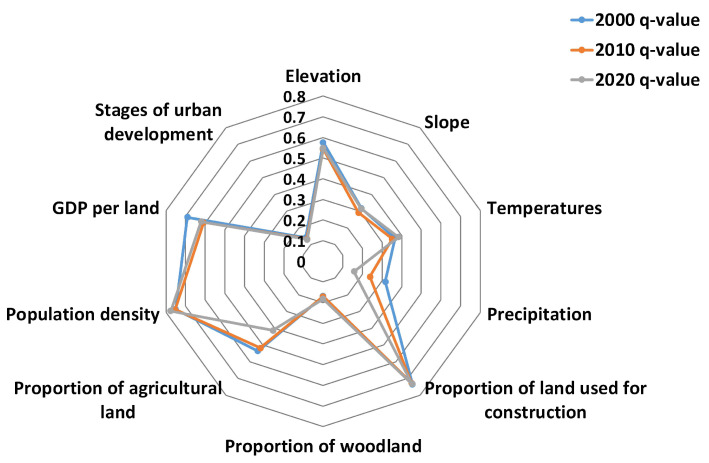
Geographical detector model factor analysis radar map.

**Figure 8 ijerph-19-16100-f008:**
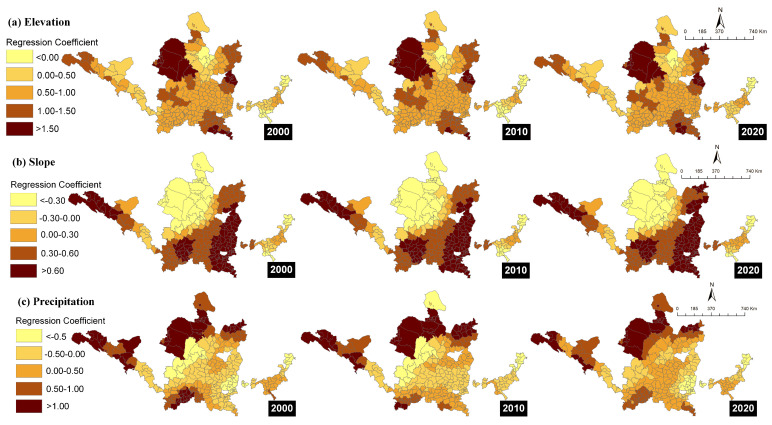
Spatial distribution of impact factor regression coefficients from 2000–2020.

**Figure 9 ijerph-19-16100-f009:**
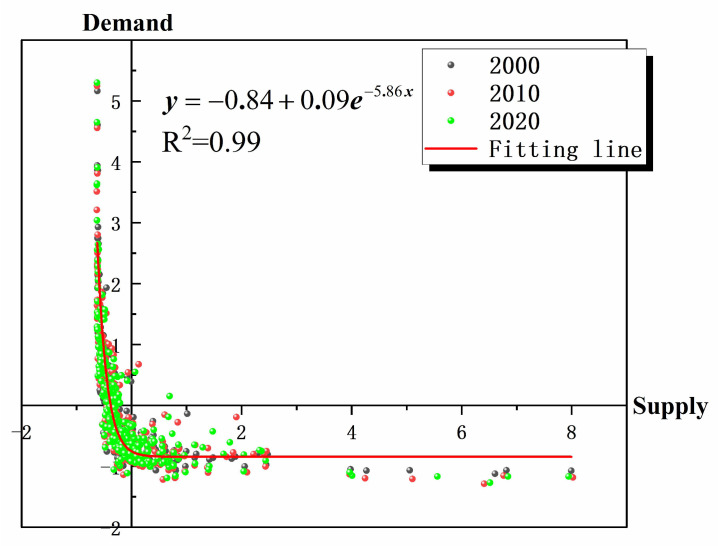
Nonlinear curve fitting of supply and demand.

**Table 1 ijerph-19-16100-t001:** The equivalence coefficients of ESs per unit area.

Type I	Type II	Agricultural	Woodland	Grass Land	Wetland	Water	Construction Land	Unutilized Land
Provision services	Food production	0.79	0.29	0.35	0.47	0.74	0.00	0.01
	Raw materials	0.37	0.66	0.52	0.47	0.21	0.00	0.03
	Water supply	0.02	0.34	0.29	2.41	7.71	0.00	0.02
Regulating services	Atmospheric regulation	0.62	2.19	1.83	1.77	0.72	0.02	0.10
	Climate regulation	0.33	6.54	4.85	3.35	2.13	0.00	0.09
	Waste treatment	0.09	1.85	1.60	3.35	5.16	0.09	0.29
	Water regulation	0.25	3.26	3.55	22.53	95.08	0.03	0.20
Support services	Soil formation	0.96	2.66	2.23	2.15	0.86	0.02	0.12
	Nutrient cycling	0.11	0.20	0.17	0.17	0.07	0.00	0.01
	Biodiversity	0.12	2.42	2.03	7.32	2.37	0.02	0.11
Cultural services	Recreation	0.06	1.06	0.89	4.40	1.76	0.01	0.05

**Table 2 ijerph-19-16100-t002:** Parameter of the degree of land-use of different types (*P_i_*).

Type	Agricultural	Woodland	Grassland	Wetland	Water	Construction Land	Unutilized Land
*P_i_*	0.545	0.114	0.215	0.120	0.120	0.936	0.063

**Table 3 ijerph-19-16100-t003:** ESs supply in resource-based cities in the YRB from 2000 to 2020 (billion CNY).

Type I	Type II	2000	2010	2020	Rate of Change 2000–2010	Rate of Change 2010–2020	Rate of Change 2000–2020
Provision services	Food production	237.08	298.60	323.01	25.95%	8.17%	36.24%
	Raw materials	221.02	280.20	304.54	26.78%	8.69%	37.79%
	Water supply	111.14	150.38	168.24	35.31%	11.88%	51.37%
Regulating services	Atmospheric regulation	641.59	816.23	888.39	27.22%	8.84%	38.46%
	Climate regulation	1509.19	1928.08	2104.55	27.76%	9.15%	39.44%
	Waste treatment	497.60	640.70	701.75	28.76%	9.53%	41.03%
	Water regulation	1275.62	1734.90	1941.11	36.00%	11.89%	52.17%
Support services	Soil formation	818.54	1040.22	1131.54	27.08%	8.78%	38.24%
	Nutrient cycling	68.99	87.48	95.04	26.80%	8.64%	37.76%
	Biodiversity	620.04	792.01	864.49	27.74%	9.15%	39.42%
Cultural services	Recreation	277.25	354.58	387.44	27.89%	9.27%	39.75%
Total		6278.07	8123.39	8910.10	29.39%	9.68%	41.92%

**Table 4 ijerph-19-16100-t004:** Explanatory power of natural and socioeconomic factors influencing resource-based cities in the YRB from 2000-2020.

Category	Influencing Factors	2000 q-Value	2010 q-Value	2020 q-Value	Average
Natural factors	Elevation	0.575534	0.544357	0.549024	0.556305
	Slope	0.316982	0.292234	0.314731	0.307982
	Temperature	0.365293	0.352226	0.383257	0.366925
	Precipitation	0.316975	0.238479	0.157708	0.237721
Socioeconomic factors	Proportion of land used for construction	0.736342	0.728005	0.733867	0.732738
	Proportion of woodland	0.169143	0.169610	0.182215	0.173656
	Proportion of agricultural land	0.536538	0.517546	0.413186	0.489090
	Population density	0.750288	0.753995	0.776130	0.760138
	GDP per land	0.690186	0.607622	0.621392	0.639733
	Stages of city development	0.143244	0.134165	0.131011	0.136140

**Table 5 ijerph-19-16100-t005:** Results of interaction detection of spatially divergent drivers of ES supply-demand in resource-based cities in the YRB.

**2000**		A1	A2	A3	A4	A5	A6	A7	A8	A9	A10
	A1	0.575534									
	A2	0.754543	0.316982								
	A3	0.689806	0.707413	0.365293							
	A4	0.690502	0.710756	0.508002	0.316975						
	A5	0.796476	0.820116	0.777927	0.80363	0.736342					
	A6	0.695479	0.432563	0.638523	0.646208	0.793154	0.169143				
	A7	0.739633	0.718505	0.733484	0.744112	0.841892	0.594275	0.536538			
	A8	0.782527	0.802881	0.776257	0.788987	0.804387	0.803191	0.833797	0.750288		
	A9	0.768248	0.788615	0.734809	0.76178	0.781359	0.787513	0.858857	0.781846	0.690186	
	A10	0.67397	0.516187	0.488486	0.443625	0.766267	0.383585	0.651732	0.779639	0.723287	0.143244
**2010**	A1	0.544357									
	A2	0.726861	0.292234								
	A3	0.675642	0.706659	0.352226							
	A4	0.637486	0.659464	0.518382	0.238479						
	A5	0.807429	0.819385	0.773957	0.797406	0.728005					
	A6	0.695053	0.401787	0.656635	0.591284	0.778786	0.16961				
	A7	0.707963	0.754875	0.731398	0.671008	0.856253	0.607548	0.517546			
	A8	0.784777	0.807395	0.782159	0.799638	0.802721	0.801591	0.843039	0.753995		
	A9	0.72292	0.718796	0.709318	0.68439	0.757313	0.685556	0.845492	0.772404	0.607622	
	A10	0.650068	0.490644	0.457165	0.40381	0.75033	0.376488	0.628765	0.774333	0.659094	0.134165
**2020**	A1	0.549024									
	A2	0.734087	0.314731								
	A3	0.675127	0.6997	0.383257							
	A4	0.689499	0.661865	0.585935	0.157708						
	A5	0.802331	0.830999	0.801766	0.800205	0.733867					
	A6	0.70767	0.406069	0.670507	0.520478	0.795058	0.182215				
	A7	0.691792	0.645705	0.664319	0.617968	0.870971	0.536104	0.413186			
	A8	0.799943	0.846406	0.821123	0.80825	0.811653	0.822848	0.85116	0.77613		
	A9	0.746959	0.765498	0.725274	0.694272	0.761749	0.72034	0.849247	0.792613	0.621392	
	A10	0.651267	0.504976	0.479159	0.363091	0.759484	0.375183	0.557117	0.79601	0.671374	0.131011

Here, A1—elevation, A2—slope, A3—temperature, A4—precipitation, A5—proportion of land used for construction, A6—proportion of woodland, A7—proportion of agricultural land, A8—population density, A9—GDP per land, and A10—stages of city development.

**Table 6 ijerph-19-16100-t006:** Significance of influencing factors according to VIF.

Influencing Factors	*p*	VIF
Elevation	0.000784 *	5.911145
Slope	0.000012 *	3.578279
Temperature	0.132179	3.454309
Precipitation	0.000062 *	3.320045
Proportion of land used for construction	0.000000 *	1.909154
Proportion of woodland	0.000165 *	3.698159
Proportion of agricultural land	0.000000 *	2.458041
Population density	0.000000 *	1.094348
GDP per land	0.000000 *	1.061109
Stages of city development	0.021915	1.138388

Note—* significant at *p* < 0.01.

**Table 7 ijerph-19-16100-t007:** Geographically weighted regression (GWR) model evaluation.

Influencing Factors		2000		2010		2020
	*R* ^2^	AICc	*R* ^2^	AICc	*R* ^2^	AICc
Elevation	0.753	541.491	0.742	554.581	0.742	555.078
Slope	0.791	492.719	0.786	499.372	0.785	501.545
Precipitation	0.641	656.489	0.642	655.731	0.637	661.863
Proportion of land used for construction	0.853	379.632	0.866	349.284	0.894	278.693
Proportion of woodland	0.667	592.233	0.666	594.933	0.656	603.732
Proportion of agricultural land	0.715	596.396	0.697	616.428	0.674	638.505
Population density	0.823	433.962	0.808	460.283	0.806	464.713
GDP per land	0.467	697.218	0.524	668.652	0.520	668.901

## Data Availability

Data sharing not applicable.

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
