# Peer review of "Spatial-Temporal Patterns of Ecosystem Services Supply-Demand and Influencing Factors: A Case Study of Resource-Based Cities in the Yellow River Basin, China"

_ijerph, 2022, doi:10.3390/ijerph192316100_

Round 1

Reviewer 1 Report

This study measures the Ecosystem Services (LESs) supply and demand and detects where the ESs supply and demand are balanced in the YRB area. In addition, the paper used the GWR model to study spatial heterogeneity in the relationship between ESs supply and demand. Investigating the relationship between supply and demand for ESs and identifying their heterogeneity is important for developing more targeted and viable local ecosystem management strategies.

My comments are as follows: I wonder why the high-value areas of the regression coefficient, as shown in Figure 8, gradually expanded from east to west over time. How did the Socioeconomic factors change from 2000 to 2020? 

Minor comments are as follows: 

Figures should be displayed in higher resolution. 

There is lack of parenthesis for bata i in Eq.(10).

Author Response

Review 1

We really appreciate for your patience and comments, which help us greatly promote our manuscript. According to your comments, the manuscript has been carefully revised point by point, and all revisions were marked in red in the revised manuscript. The response to each comment is listed as follows.

Comments and Suggestions for Authors

This study measures the Ecosystem Services (ESs) supply and demand and detects where the ESs supply and demand are balanced in the YRB area. In addition, the paper used the GWR model to study spatial heterogeneity in the relationship between ESs supply and demand. Investigating the relationship between supply and demand for ESs and identifying their heterogeneity is important for developing more targeted and viable local ecosystem management strategies.

My comments are as follows:

Q1(R1). I wonder why the high-value areas of the regression coefficient, as shown in Figure 8, gradually expanded from east to west over time. How did the Socioeconomic factors change from 2000 to 2020?

Response:

Thank you for your valuable comment. In 1996, China 's economic development strategy put forward the policy of supporting the developed areas in the east to support the backward areas in the west, so as to bring capital investment, talent investment, industrial investment and other resources to the western region with backward economic foundation, and improve the infrastructure construction and industrial development mode in the western region. This policy has led to an increase in the rate of economic development in the west, an increase in the area of land used for construction and a consequent increase in population. At the same time, previous studies have shown that cities in the central (Shanxi, Shaanxi) and eastern regions (Shandong) have stronger economic diffusion capacity[1]. The advantages of cities in resources, technology and labor force should be systematically infiltrated into the surrounding areas and larger areas to improve the economic development level and ability of these areas. From this point of view, the high-value area of the regression coefficient of the surrounding cities in the eastern region will also expand to the surrounding and western regions.

The discussion section has also been modified accordingly, please refer Lines 565-571,Lines 605-613 and Line 646-650.

At the same time, previous studies have shown that cities in the central (Shanxi, Shaanxi) and eastern regions (Shandong) have stronger economic diffusion capacity. The advantages of cities in resources, technology and labor force should be systematically infiltrated into the surrounding areas and larger areas to improve the economic development level and ability of these areas, thus increasing ESs demand.

However, China 's economic development strategy put forward the policy of supporting the developed areas in the east to support the backward areas in the west, so as to bring capital investment, talent investment, industrial investment and other resources to the western region with backward economic foundation, and improve the infrastructure construction and industrial development mode in the western region. This policy has led to an increase in the rate of economic development in the west, an increase in the area of land used for construction and a consequent increase in population. This has caused the area of high regression coefficients for economic factors to expand from east to west over time.

In terms of spatial distribution, the absolute values of the regression coefficients of socioeconomic impact factors such as GDP, population density, and the proportion of land used for construction were greater in the eastern YRB than in the western YRB, but the high-value areas of socio-economic impact factors tend to expand to the west.

Q2(R2). Figures should be displayed in higher resolution.

Response:

We thank the reviewer for this suggestion. The resolution of the previous part of the picture is 300, and the resolution of all the pictures has been adjusted to 1200 and replaced in the original manuscript.

Q3(R3). There is lack of parenthesis for bata i in Eq.(10).

Response:

We thank the reviewer for this suggestion. The formula in the original text is in line 265.

References

  1. L. Zhang, X. Yang. The Correlation Analysis of Urban Economic Diffusion Effect and Rural Revitalization Development Level in China. Statistical Theory and Practice 2022, doi:10.13999/j.cnki.tjllysj.2022.06.00636-42.

Reviewer 2 Report

The manuscript presents original and multi-criteria research on the spatial-temporal patterns of ecosystem services supply demand and influencing factors in the Yellow River Basin, China. 

The design of the research, the methods and objectivity do not raise my objections. However, I would ask for some clarifications to be made:

- Some shortcuts (e.g., GDP) are not explained in the text.

- Regarding the equivalence coefficients of ESs per unit area - the text explains how those related to the provision services were calculated. And how were the others obtained? (Table 1)

- Table 2 - it is unclear how the Pi parameter was obtained (from the data source?)

- The figures are pixelized, which reduces their legibility.

Author Response

Review 2

We really appreciate for your patience and comments, which help us greatly promote our manuscript. According to your comments, the manuscript has been carefully revised point by point, and all revisions were marked in red in the revised manuscript. The response to each comment is listed as follows.

Comments and Suggestions for Authors

The manuscript presents original and multi-criteria research on the spatial-temporal patterns of ecosystem services supply demand and influencing factors in the Yellow River Basin, China.

The design of the research, the methods and objectivity do not raise my objections. However, I would ask for some clarifications to be made:

Q1(R1). Some shortcuts (e.g., GDP) are not explained in the text.

Response:

We thank the reviewer for this suggestion. The first place where the abbreviation appears should be explained, at the same time we check the full text and found other similar problems, we have modified the sentence as follows, please refer Lines 27-29,Line 269-271 and Line 482-483.

GDP: Gross Domestic Product.

Line 27-29, “Moreover, the relationship between supply and demand was significantly affected by natural and socioeconomic factors, such as elevation, slope, precipitation, land-use type, population density, and Gross Domestic Product (GDP) per land.”

AIC: Akaike Information Criterion.

Line 269-271, “In this study, the spatial weight matrix is determined by a Gaussian function, while the optimal bandwidth is obtained by the Akaike Information Criterion (AIC) method”

OLS: Ordinary Least Squares.

Line 482-483, “In this study, the eight indicators screened by Ordinary Least Squares (OLS) were regressed using the GWR model”

Q2(R2). Regarding the equivalence coefficients of ESs per unit area - the text explains how those related to the provision services were calculated. And how were the others obtained? (Table 1)

Response:

We thank the reviewer for this comment. The equivalence coefficients table mentioned in the problem is modified on the basis of the equivalence coefficients table proposed by Gaodi Xie[1]. At the same time, the equivalence coefficient needs to be corrected when analyzing the study area. The formula of the correction coefficient is as follows,

λ=Pstudy/Ptotal

here,  λ is correction coefficient; Pstudy is the grain production per unit area in the study area; Ptotal is the grain production per unit area nationwide. Because the study area involves 9 provinces, the average grain production per unit area of these 9 provinces is taken as the grain production per unit area of the study area. According to this ratio, the correction coefficient is 0.93. Therefore, in the equivalence coefficients table, this value is directly used to correct all equivalence coefficients. The data used in the calculation are as follows (Table 1),

Table 1 The grain production per unit area of study area

Area

Shanxi

Inner Mongolia

Shaanxi

Gansu

Henan

Shandong

Ningxia

Sichuan

Average value

China

Production (kg·km-1·a-1)

4550

5362

4238

4557

6356

6577

5602

5588

5355

5734

In the original text, the scope of application of this correction coefficient is also explained. The revised content is shown in line 165-166:

The value of ESs equivalents in the YRB was determined using a correction factor of 0.93 according to the ratio of grain production per unit area in the study area to the grain production per unit area nationwide, this correction factor was used to calculate the equivalence coefficients of all services.

Q3(R3). Table 2 - it is unclear how the Pi parameter was obtained (from the data source?)

Response:

We thank the reviewer for this suggestion. This value is obtained by Hebing Hu[2] combining the expert scoring method and the Leopold matrix method, it’s a method to determine the parameter of land use degree according to land use type, which is essentially similar to the grading assignment method proposed by Dafang Zhuang[3] (Table 2). However, in comparison, the Hu’s classification is more accurate and the grading is more scientific.

Table 2 The classification values of land use degree (Dafang Zhuang)

Type

Unused land

Forest, grass, water land level

Agricultural land level

Land Level for Urban Settlements

Land use type

Unused land

Woodland, grassland, waters

Cultivated land, garden land, artificial grassland

Towns, settlements, industrial and mining land, transportation land

Land use degree

1

2

3

4

This situation is also explained in the original manuscript. The revised content is shown in line 191-192:

“and Pi is the parameter of the degree of land use of different types, this value is obtained by the predecessors [38,39] combining expert scoring method and matrix method (Table 2).”

Q4(R4). The figures are pixelized, which reduces their legibility.

Response:

We thank the reviewer for this suggestion. The resolution of the previous part of the picture is 300, and the resolution of all the pictures has been adjusted to 1200 and replaced in the original manuscript.

References

  1. G. Xie; Y. Xiao, L. Zhen. Study on ecosystem services value of food production in China. Zhongguo Shengtai Nongye Xuebao 2005, 13, 10-13.
  2. H. Hu; H. Liu; J. Hao, J. An. Spatio-temporal variation in the value of ecosystem services and its response to land use intensity in an urbanized watershed. Acta Ecologica Sinica 2013, 33, 2565-2576.
  3. D. Zhuang, J. Liu. Study on the Model of Regional DIifferentiation of Land Use Degree in China. Journal of Natural Resources 1997, 10-16.